# Spatiotemporal Changes in the Hydrochemical Characteristics and the Assessment of Groundwater Suitability for Drinking and Irrigation in the Mornag Coastal Region, Northeastern Tunisia

**Emna Hfaiedh** [1,2,3], **Amor Ben Moussa** [1], **Marco Petitta** [2,*] and **Ammar Mlayah** [3]

1   Research Laboratory of Environmental Science and Technologies, High Institute of Environmental Science and Technology (HIEST) of Borj Cedria, University of Carthage, Amilcar 1054, Tunisia; emna.hfaiedh@uniroma1.it (E.H.); amor_geologie@yahoo.fr (A.B.M.)
2   Earth Sciences Department, Sapienza University of Rome, 00185 Rome, Italy
3   Centre of Water Research and Technologies, University of Carthage, Amilcar 1054, Tunisia; ammarmlayah17@gmail.com
*   Correspondence: marco.petitta@uniroma1.it

**Abstract:** Hydrogeochemical properties and groundwater quality assessment are very important for the effective management of water resources in arid and semiarid regions. The present investigation is a spatiotemporal analysis of groundwater quality using both chemical analysis and water quality indices (WQIs) in the Mornag Basin in northeastern Tunisia. The results exhibit that the Mornag shallow aquifer is dominated by chloride–sodium–potassium water facies, which progress over time toward chloride–sulfate–calcium and magnesium water facies. This may highlight that the mineralization of groundwater, which increases in the direction of groundwater flow, is primarily controlled by a natural process resulting from the dissolution of evaporative minerals and cation exchange with clay minerals relatively abundant in the study area. The anthropogenic activities represented by the return flow of irrigation water, the recharge by wastewater, and climate change also have a key role in groundwater contamination. The temporal evolution in %Na and SAR over the last three years in the Mornag aquifer shows an increasing trend that makes them unsuitable for irrigation. These findings highlight the need for assessing water quality in mapping local water resource vulnerability to pollution and developing sustainable water resources management.

**Keywords:** hydrogeochemical; spatiotemporal evolution; Mornag shallow aquifer; WQI; unsuitability





## 1. Introduction

In the semi-arid and arid areas of North Africa, groundwater resources play an extremely important role in socio-economic development, restoring natural ecosystems, and mitigating climate change [1,2]. Under these climatic conditions, the southern regions of the Mediterranean (from Morocco to Egypt) are distinguished by relatively limited groundwater resources and an uneven distribution in space and time. Despite the excessive use of these resources, the demand for water will continue to rise as a result of population growth, the expansion of irrigated areas, industry, and tourism. It is crucial to consider the limitations of the current synergies between protecting coastal areas, combating climate change, anthropization, and adapting to its effects. In addition, improving water governance is, without doubt, a key challenge for future regional progress. In fact, the chemical ion content resulting from the water–mineral interaction of the surrounding rock, or from anthropogenic activities, may directly impact water quality [3–8]. The resultant physicochemical composition of the groundwater can have a significant impact on human health, quality of life, and cultural growth. Indeed, human activities related to industrialization, agricultural development, and urbanization have introduced high levels of chemical elements into water resources, causing a deterioration in water quality [9–11].

Therefore, to ensure water safety in dry conditions, water contamination must be prevented and mitigated. In this regard, we are interested in the spatiotemporal evolution of the groundwater mineralization of the Mornag shallow aquifer. The study area of Mornag, which is located 20 km south of the city of Tunis with a total area of approximately 370 km$^2$, has a Mediterranean climate characterized by irregular rainfall and hot temperatures [2,12]. The average monthly temperature is around 22 °C, with annual precipitation ranging from 292 to 800 mm/year. In recent years, extensive irrigation and industrial development have taken place in this semi-arid region, contributing not only to increased agricultural production and economic growth but also unfortunately to a decline in the water table and groundwater quality degradation by the return flow of irrigation water and the recharge caused by wastewater [2].

Understanding the spatiotemporal variations in the mineralization and the major sources of pollution is an effective means of mitigating and controlling the continued deterioration in groundwater quality. In fact, such monitoring can help create databases whose analysis can be used as a decision-making tool for the sustainable management of water resources.

In the present study, which uses a combination of physicochemical data and water quality indices, a special emphasis is placed on the monitoring and evaluation of spatial and temporal changes in groundwater quality tracked for three years (2013, 2016, and 2019) and on the identification of sources of salinization of groundwater.

## 2. Materials and Methods

### 2.1. Study Area Description

Geologically, the Mornag graben seems to be a large basin, controlled by tectonic events from N-S (Figure 1b) and NW-SE to N160, which have affected the Mio-Plio-Quaternary filling [12]. The Mornag Basin is characterized by the predominance of heterogeneous quaternary sediments, consisting predominantly of conglomerates, sand, gravel, silt, and clay. These deposits are bounded to the northwest by the late Miocene sediments, to the southeast by the paleogenic sediments, and to the east by ancient sediments ranging from the Triassic to late Eocene [13–16] (Figure 1a). In the eastern part, particularly in Jebel Ressas, there are deposits belonging to the late Cretaceous-Triassic interval, which are composed of marl with thin limestone lenses. Paleocene sediments are found in the south and southeast parts of the basin. The eastern part of Jebel Ressas and the western part of Jebel Boukornine are distinguished by Eocene deposits. Oligocene sediments are found in the south and west-central regions. However, Miocene deposits, which occur in the northwestern part of the study area, consist of gypsum, clay, limestone, marl, and sandstone [16–20].

The geological formations in the Mornag graben host a complex multi-layered aquifer system [13,16,21–23]. The shallow aquifer, which is unconfined throughout the study area, consists mainly of quaternary alluvial deposits. According to both [23] and [24], this shallow aquifer has a storage coefficient of $6 \times 10^{-3}$ to $7.3 \times 10^{-5}$ and a transmissivity of 1 to $80 \times 10^{-4}$ m$^2$/s. The recharge occurs mainly by the infiltration of rainfall at the foot of the surrounding hills, specifically in the regions of Khlédia, Ettella, and Rorouf [25]. However, it should be noted that this shallow aquifer has been affected by anthropogenic pressures, which are associated with the growing demand for the agriculture sector, the major activity in the area [2,12].

The deep aquifers are logged in deposits that extend from the upper Eocene to the Quaternary, with a storage coefficient and transmissivity ranging from $5 \times 10^{-3}$ to $6 \times 10^{-3}$ and from $10^{-2}$ to $10^{-4}$ m$^2$/s, respectively [21,24]. This deep aquifer is separated from the shallow one by thin impermeable layers of a clayey nature. However, there are vertical communications between the superimposed hydrogeological units, which have been facilitated by the lateral and vertical discontinuity of the lithological facies and the difference in charge between the aquifer levels [16,26].

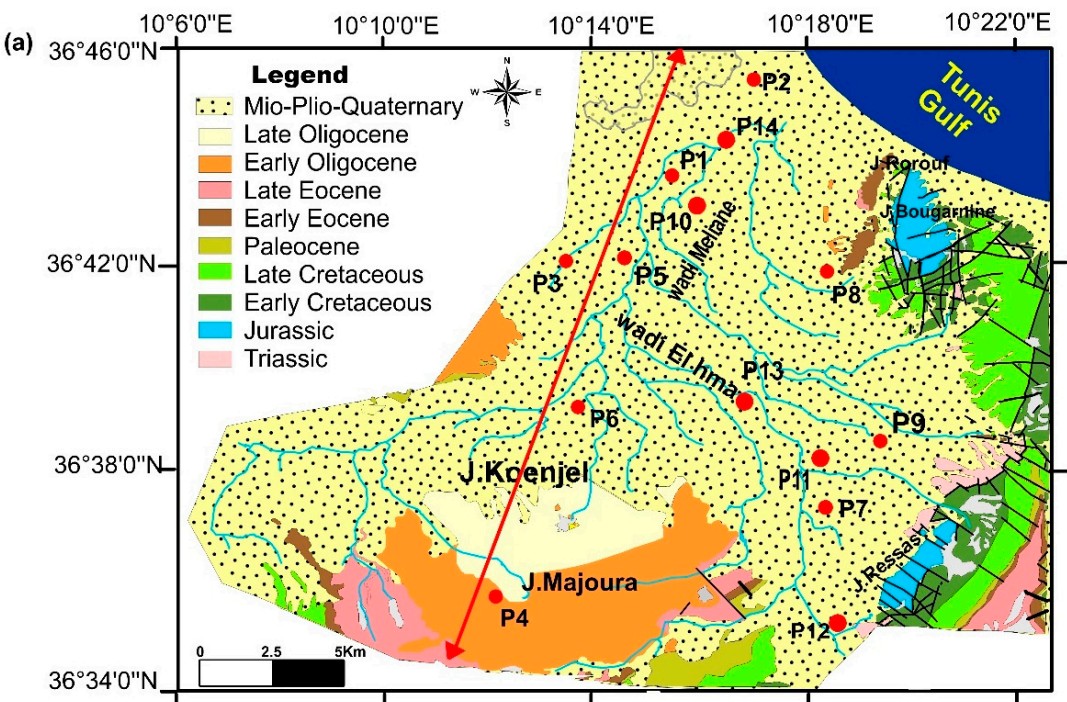

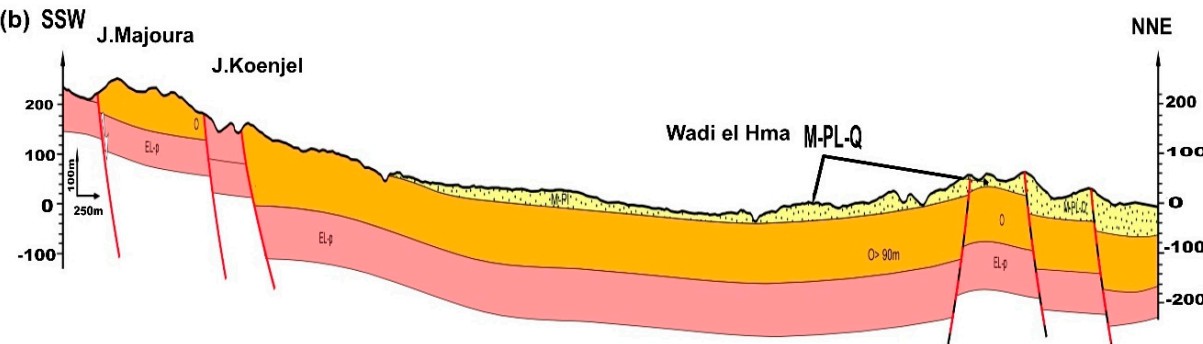

**Figure 1.** (**a**) Location and sampling map of the study area [14,15] (**b**) simplified geological cross-section of the study area.

The drainage network consists of the Meliane principal wadi and its tributary represented by the El Hma wadi, which merges to the northwest of the plain towards the Gulf of Tunis. From a hydrological point of view, the study area is divided into three sub-watersheds, i.e., the Oued Meliane basin, the Oued El Hma basin, and the Medjerda Cap-Bon channel. The latter corresponds to the downstream part of the great watershed of Oued Méliane, where the natural outlet is the Mediterranean Sea [23,27].

Soil types in the study district fall into the category of calcareous brown calcimagnesic soils and rendzines [28,29]. The texture of these soils, which are shallow, often steeply sloping, is sandy to silty in the western and southwestern portions of the plain and silty to sandy clay in the eastern part [30]. However, in the central and northern parts, soils belong to a class that is not highly reactive because of alluvial deposits and have large areas of vertic soils. These are deep, with clay to silty texture, and are moderately adapted to irrigation. The soils of the southern and southwest parts of the study area belong to the class of Mediterranean soils and brown steppes. They are typically deep, with a light texture of sandy to silty and sandy clay to silty (Figure 2).

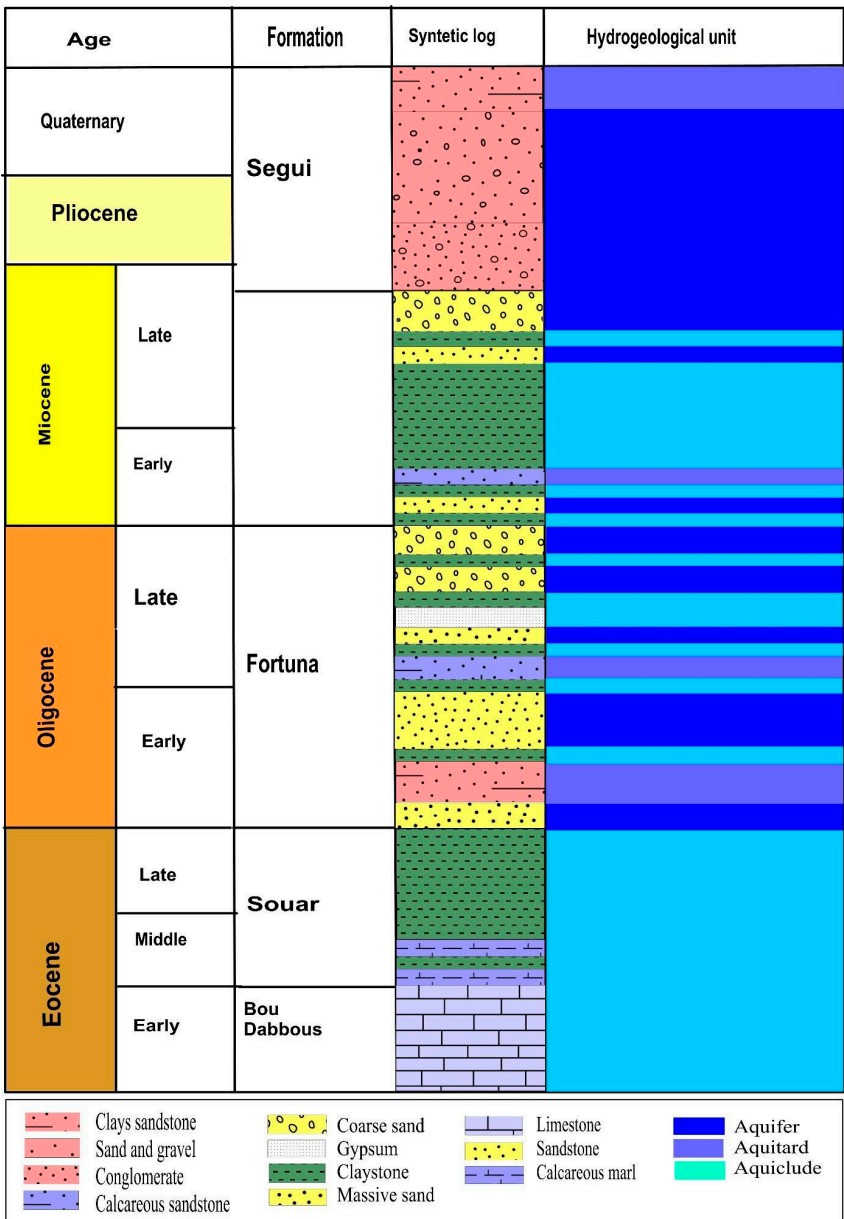

**Figure 2.** Log hydrostratigraphic of the study area.

The potentiometric map (Figure 3) shows Mornag's shallow water table flowing from the southwest to the northeast of the basin, from the foothills adjacent to the Gulf of Tunis corresponding to the discharge zone. The main recharge areas are situated at the foot of the Tella and Rorouf mountains, as well as the Bir El Kasaa and Rades-Megrine hills, which are the secondary feeding areas. Piezometric levels measured in 2008 (Figure 3a) and 2018 (Figure 3b) vary widely between 5 m and 60 m a.s.l. This may indicate a gradual decline in the piezometric surface downstream of the plain near the coast. The relatively low piezometric levels that are measured in the areas of Zaouiet Mornag, Bou Mhel, and Sebakhet Mornag demonstrate the effect of over-exploitation of the shallow water table. Examination of these two maps shows that a significant increase in the piezometric surface was observed during the 2018 campaign, particularly on the foothills of the surrounding mountains. This is likely due to the considerable amount of precipitation that occurred during this year's wet period. In addition, this rise in the piezometric level can be explained by the impact of the operations of artificial recharge by the waters of the North, precisely in the Khlédia region.

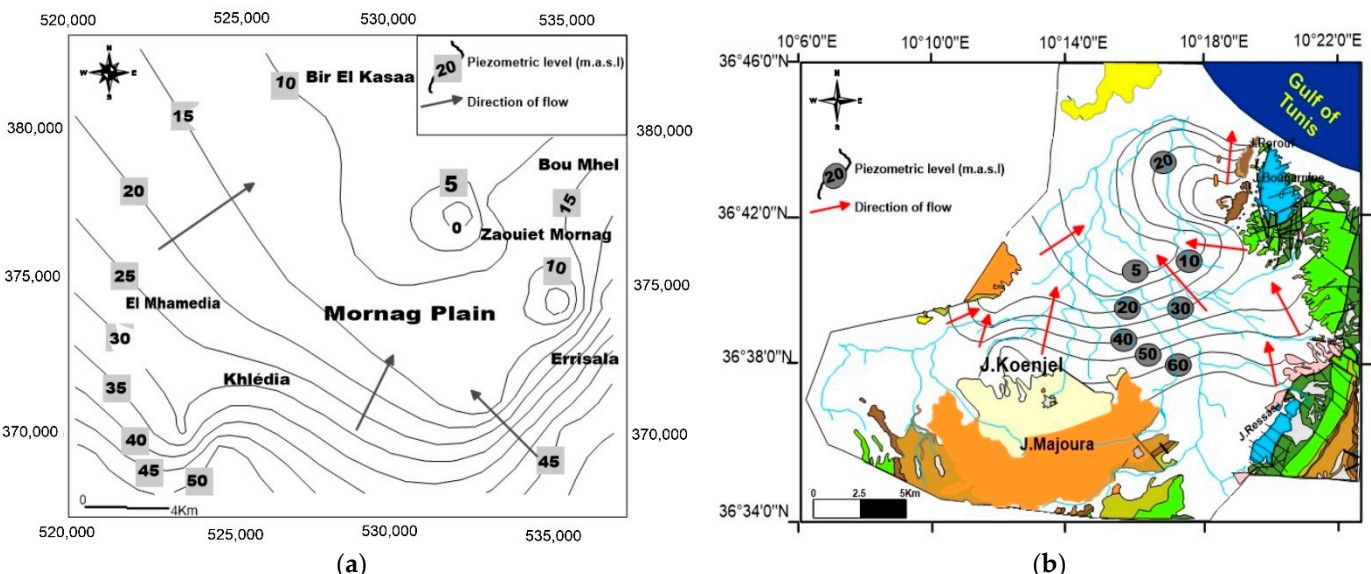

**Figure 3.** Piezometric maps: (**a**) piezometric maps from 2008 [12]; (**b**) piezometric maps from 2018 [31].

### 2.2. Sampling and Data Treatment

In this study, physicochemical analyses, such as pH, temperature, and electrical conductivity (EC), were carried out in situ. The concentrations of major ions, i.e., chloride (Cl), calcium (Ca), magnesium (Mg), sodium (Na), potassium (K), sulfate (SO$_4$), and bicarbonate (HCO$_3$), were conducted in the Environment Science and Technology Research Laboratory. The PHREEQ geochemical program [32] was used to calculate the waterless speciation and the thermodynamic equilibrium conditions (mineral saturation indices) of the waters with respect to the major mineral phases present in the aquifer. The saturation index (*SI*) quantitatively describes the difference between water and balance with respect to dissolved minerals and is given by the equation below [33,34]:

$$SI = log\left(\frac{IAP}{Kt}\right) \tag{1}$$

where *IAP* is the ion exertion product and *Kt* is the equilibrium solubility constant.

To study the hydrochemical characteristics of the Mornag shallow aquifer groundwater and identify the various types of water, Diagrammes software was used to develop the Piper diagram for Mornag groundwater corresponding to the years 2013, 2016, and 2019 [35].

This software was also used to evaluate the quality of irrigation water in the Mornag study area by calculating the risk of alkalinity hazard (*SAR*) [36] and the percentage of sodium (%Na) [37].

$$SAR = \left(\frac{Na^+}{\sqrt{\left(Ca^{2+} + Mg^{2+}\right)/2}}\right) \times 100 \text{ meq/L} \tag{2}$$

$$Na\% = \frac{\left(Na^+ + K^+\right)}{\left(Ca^{2+} + Mg^{2+}\right) + \left(Na^+ + K^+\right)} \times 100 \tag{3}$$

Principal component analysis (PCA) was performed using XLSTAT software, which covered 9 variables (pH, TDS, Na$^+$, Ca$^{2+}$, Mg$^{2+}$, K$^+$, SO$_4$$^{2-}$, Cl$^-$, and HCO$_3$$^-$) [38].

## 3. Results and Discussion

### 3.1. Water Facies

To identify the chemical facies, the results of the groundwater major ion analysis were plotted on a Piper diagram (Figure 4) [35]. The Mornag shallow groundwater samples were classified into two major hydrochemical facies: the Na-K-Cl water type, which has evolved over time to the Ca-$SO_4$-Cl water type. Changes in chemical facies from upstream to downstream can be linked to the geological nature of the aquifer, human activities, and/or climate changes over time.

### 3.2. Hydrochemical Characteristics

The chemistry of the groundwater samples in 2013, 2016, and 2019 are listed in Table 1. These values indicate that the dominant cation was sodium during the three years and that the dominant anion was chloride, giving water and drinks a salty taste [39].

The relative order of abundance of the major cations $Ca^{2+}$, $Mg^{2+}$, $Na^+$, and $K^+$ with an average of 38%, 13%, 48%, and 1% of all cations, respectively, for $HCO_3^-$, $Cl^-$, and $SO_4^-$ anions account for an average of 15%, 49%, and 36% of the total anions, respectively (for the year 2013). The measurements established in 2016 show an abundance of $Ca^{2+}$, $Mg^{2+}$, $Na^+$, and $K^+$ cations of 30%, 11%, 57%, and 2% of the total cations, respectively, and $HCO_3^-$, $Cl^-$, and $SO_4^{2-}$ anions account for an average of 30%, 43%, and 27% of the total anions, respectively. Nonetheless, the chemical composition measured in 2019 shows an abundance of $Ca^{2+}$, $Mg^{2+}$, $Na^+$, and $K^+$ cations with an average of 28%, 14%, 56%, and 2% of the total cations, respectively, and $HCO_3^-$, $Cl^-$, and $SO_4^-$ anions represent an average of 27%, 38%, and 35% of all anions, respectively. Cation abundance shows a decrease in calcium concentrations over time between 2013 and 2019, with higher levels detected at well P6 in 2013 and increasing over time. Regarding magnesium, sodium, and potassium, higher concentrations were found in wells P2, P8, and P1 in 2019. These higher concentrations can be related to water–rock interaction processes, such as the dissolution of carbonate and evaporative minerals and cation exchange. High levels of potassium may be due to the excessive use of potassium-based chemical fertilizers and soil leaching after fertilizer use.

Analyses of anions show a decrease in chloride concentration over time with a higher concentration in 2013 at well P4. Indeed, an increase in bicarbonate and sulfate levels was detected over time in P14 in 2019. These ion concentrations are mainly controlled by natural and/or anthropogenic mineralization processes related to water–rock interactions, human activities, and climate change.

### 3.3. Statistical Analysis

To investigate the relationships between major elements and to identify their origin in the Mornag groundwater, principal component analysis (PCA) was conducted on a data set of 14 samples and 9 physicochemical parameters, i.e., pH, TDS, $Na^+$, $Ca^{2+}$, $Mg^{2+}$, $K^+$, $SO_4^{2-}$, $Cl^-$, and $HCO_3^-$ (Table 1). Each factor is defined by a number of variables that control particular mechanisms of water mineralization [38,40].

The two preserved factors account for the maximum total variance of the samples. F1 and F2 were subsequently calculated for the individual samples and used for determining the position of the intensity of each of the factors controlling the hydrochemistry in each position.

The first factor, recording about 55%, 70%, and 57% of the total variance in 2013, 2016, and 2019, respectively, is, in large part, correlated with salinity, Na, Mg, Ca, $SO_4$, and Cl. The second factor, accounting for 16%, 15%, and 23% of the total variance in 2013, 2016, and 2019 is highly associated with $HCO_3$, K, and pH (Figure 5).

Table 1. Hydrogeochemical data of Mornag shallow groundwater (mg/L).

| | 2013 | | | | | | | 2016 | | | | | | | 2019 | | | | | | |
|---|---|---|---|---|---|---|---|---|---|---|---|---|---|---|---|---|---|---|---|---|---|
| | $Ca^{2+}$ | $Mg^{2+}$ | $Na^+$ | $K^+$ | $HCO_3^-$ | $Cl^-$ | $SO_4^-$ | $Ca^{2+}$ | $Mg^{2+}$ | $Na^+$ | $K^+$ | $HCO_3^-$ | $Cl^-$ | $SO_4^-$ | $Ca^{2+}$ | $Mg^{2+}$ | $Na^+$ | $K^+$ | $HCO_3^-$ | $Cl^-$ | $SO_4^-$ |
| P1 | 193.4 | 122.8 | 828.7 | 4.1 | 170.1 | 1225.5 | 1174.9 | 195.9 | 90.3 | 583.7 | 20.4 | 359.6 | 969.5 | 578.8 | 275 | 120.5 | 506.1 | 45.1 | 480 | 555 | 257.5 |
| P2 | 351.5 | 216.3 | 710.2 | 30.3 | 148.8 | 1446.1 | 1149.4 | 146.8 | 72.4 | 412.8 | 23.6 | 390.4 | 542.53 | 476.2 | 102.7 | 73.4 | 168.9 | 12.3 | 240 | 420 | 238 |
| P3 | 164.7 | 98.27 | 733.5 | 3.3 | 158.6 | 1024.5 | 545.5 | 97.1 | 44.8 | 217.9 | 18.6 | 488 | 235.4 | 242.6 | 199.8 | 80.4 | 585.2 | 16.3 | 160 | 555 | 210.1 |
| P4 | 159.3 | 78.74 | 220.7 | 20.9 | 109.8 | 723.9 | 65 | 481.7 | 195.8 | 476.5 | 30.8 | 616.1 | 1467.5 | 239.1 | 159.2 | 90.3 | 270.5 | 12.4 | 320 | 435 | 389.5 |
| P5 | 106.7 | 34.25 | 213.9 | 5.7 | 0 | 377.4 | 360.3 | 360.2 | 235 | 476.5 | 5.8 | 451.4 | 1368.4 | 1190.1 | 380.4 | 273 | 1482 | 18.1 | 400 | 1005 | 272.8 |
| P6 | 383.6 | 90.83 | 143.5 | 3.1 | 168.3 | 416.7 | 935.1 | 272.6 | 73.4 | 476.5 | 17.9 | 311.1 | 625 | 171.8 | 404.6 | 188.6 | 1092 | 19.2 | 360 | 885 | 637.2 |
| P7 | 199.1 | 34.15 | 129.8 | 2.6 | 145.1 | 359.5 | 93.3 | 125.9 | 32.2 | 476.5 | 3.8 | 244 | 161.9 | 61.9 | 347.1 | 188.2 | 601.3 | 13.5 | 280 | 585 | 249.1 |
| P8 | 176.7 | 46.83 | 89.6 | 4.6 | 265.9 | 241.85 | 102.8 | 268.6 | 59.6 | 476.5 | 3 | 359.9 | 583.4 | 427.5 | 1013 | 167.4 | 2788 | 13.7 | 600 | 900 | 1480.1 |
| P9 | 400.1 | 184 | 725.2 | 21.5 | 480.6 | 1996.87 | 562.8 | 107.5 | 41 | 476.5 | 9.6 | 353.8 | 233.3 | 83.9 | 335 | 203 | 623 | 17.3 | 320 | 915 | 207.4 |
| P10 | 705.5 | 183.4 | 458.8 | 6.6 | 195.2 | 1732.15 | 970.49 | 224.05 | 98.2 | 368.7 | 2 | 524.6 | 439.6 | 678.9 | 138.9 | 60.3 | 231.7 | 10.6 | 240 | 155 | 172.6 |
| P11 | 197.9 | 63.18 | 253 | 4.8 | 158.6 | 102.95 | 472.7 | 869.7 | 254 | 2838.3 | 124 | 317.2 | 5564.6 | 1621.7 | 144.1 | 64.8 | 201.8 | 11.8 | 240 | 145 | 319.85 |
| P12 | 161.3 | 71.88 | 219.3 | 3.7 | 220.82 | 482.06 | 393.7 | 256.1 | 67 | 150.8 | 3 | 347.7 | 307.4 | 499.7 | 104.5 | 89 | 362.6 | 9.5 | 280 | 140 | 345.8 |
| P13 | 438.9 | 81.93 | 260.5 | 5.09 | 201.3 | 594.81 | 836.7 | 172.8 | 58.6 | 217.4 | 3.3 | 427 | 365.2 | 248.6 | 223.3 | 69.9 | 187.3 | 7 | 320 | 111 | 625.3 |
| P14 | 118.5 | 99.95 | 428.2 | 15 | 376.9 | 768.03 | 364.9 | 348.9 | 179.4 | 872.6 | 8.1 | 475.8 | 1652.7 | 953.9 | | | | | 280 | 205 | 1285 |

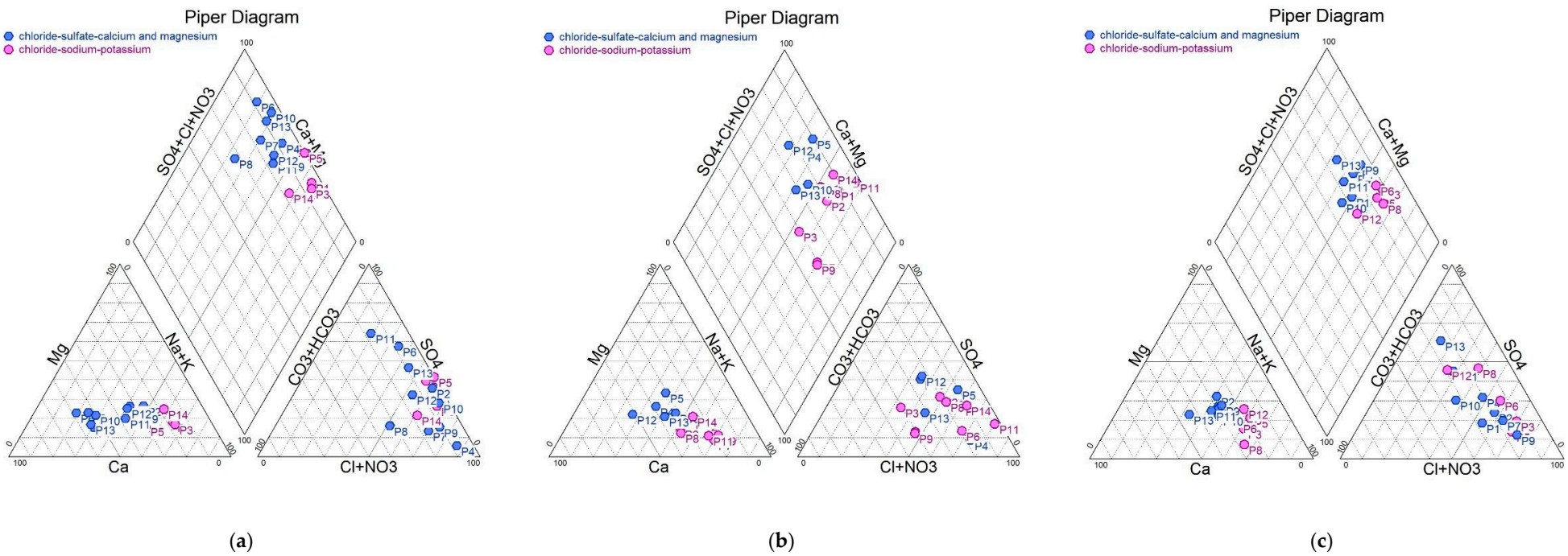

**Figure 4.** Piper diagrams (2013, 2016, 2019). (**a**) Piper diagram for 2013; (**b**) Piper diagram for 2016; (**c**) Piper diagram for 2019.

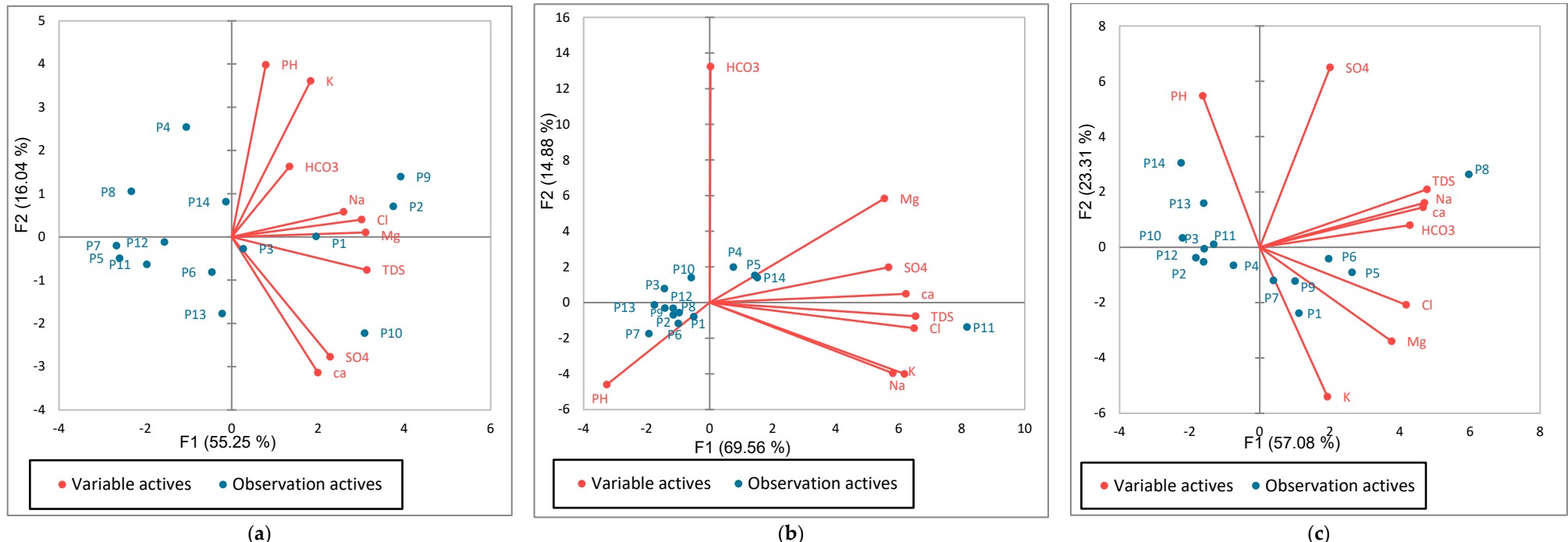

**Figure 5.** (**a**) Variable spaces and cluster analysis deduced from the geochemical PCA for 2013. (**b**) Variable spaces and cluster analysis deduced from the geochemical PCA for 2016. (**c**) Variable spaces and cluster analysis deduced from the geochemical PCA for 2019. The correlation of the samples of Mornag indicated that the contents of sodium, magnesium, and calcium are highly positively correlated with the sum of cations, and on the other hand, the anions $Cl^-$, $SO_4^{2-}$, and $HCO_3^-$ have a positive correlation with respect to the sum of the anions, with a correlation coefficient (R).

The positive total variance for the two first factors (F1 and F2), which increases generally over time, highlights that these ions continue to increase along the groundwater flow path and, thus, contribute largely to the mineralization of groundwater. Moreover, the F1 factor indicates the involvement of water–rock interaction processes, such as the dissolution of evaporative minerals (halite and gypsum). The ion behaviors expressed by factor F2 are interpreted as the result of an exchange of cations with clayey minerals, from agricultural practices to wastewater discharge [41,42].

### 3.4. Groundwater Quality and Assessment for Drinking

The pH of the water samples ranges from 6.94 to 8.34 in 2013 and from 6.82 to 8.2 in 2016. All samples are within the acceptable limits for human consumption, which vary between 6.5 and 8.5 [43]. However, for 2019, these values vary from 4 to 7.4, indicating that they are not within the permissible limits. Groundwater temperature values range from 19.1 °C to 27.8 °C in 2013, showing that some samples are off limits. Temperature measurements for 2016 and 2019 range from 18.4 °C to 24.3 °C and 17.1 °C to 21.8 °C, respectively. This may indicate that they meet the 25 °C guideline for drinking water [43]. The electrical conductivity of groundwater ranges from 1.8 to 6.7 ms/cm in 2013, from 1.2 to 11 ms/cm in 2016, and from 0.7 to 2.07 ms/cm in 2019. The TDS values range between a minimum of 928.52 mg/L and a maximum of 4052.82 mg/L in 2013, between 1106.51 and 11,589.5 mg/L in 2016, and from 1009.234 to 696.231 mg/L in 2019. This may indicate that all groundwater samples have TDS values above normalized limits, with the exception of wells 8 and 9 in 2013 [43]. According to the classifications of [44–46], groundwater in the study area is predominantly brackish and, therefore, not suitable for human consumption.

The spatial distribution of major ion concentrations displays a wide range of variation. The sodium contents vary from 89.69 to 828.75 mg/L in 2013, from 150.88 to 2838.3 mg/L in 2016, and from 168.9 to 1482 mg/L in 2019. This suggests that all samples were above the WHO standard of 200 mg/L, with the exception of wells 7, 8, and 9 in 2013; well 13 in 2016; and wells 2 and 13 in 2019. Sodium contents increased over time from 2013 to 2019 in the same direction as groundwater flow. However, in the northwestern part of the basin, groundwater samples collected from wells 1, 2, 3, 5, 10, and 14 exhibit sodium concentrations that decrease over time. Calcium concentrations that gradually increase in the direction of flow and over time vary from 106.72 mg/L to 705.57 mg/L in 2013, from 97.13 mg/L to 869.7 mg/L in 2016, and from 102.7 mg/L to 1013 mg/L in 2019 (Figure 6). The desired drinking water limit for $Ca^{2+}$ is set at 75 mg/L [47]. According to the standard guideline values, all groundwater samples are unsuitable for drinking water. These higher concentrations are mainly related to natural mineralization processes, such as the dissolution of carbonate and evaporative minerals and cation exchange [36]. Consequently, the optimal concentration of $Ca^{2+}$ prevents heart disease and supports the normal functioning of the human metabolic process.

The spatial distribution maps (Figure 6) show that $Mg^{2+}$ concentrations, which vary significantly between 34.15 and 216.3 mg/L in 2013, increase from south to north of the study area from 32.21 to 254 mg/L in 2016 and between 60.34 and 273 mg/L in 2019. The majority of groundwater samples are characterized by $Mg^{2+}$ concentrations above the WHO-proposed allowable limit (150 mg/L), which can be explained by the dissolution of dolomite and/or epsomite. They provide drinking water that exceeds the recommended limits for $Mg^{2+}$ and increases the risk of cardiovascular disease and a serious drop in blood pressure [48].

The chloride content shows an overall trend with an increase from upstream to downstream. They range from 102.95 to 1996.87 mg/L in 2013 and from 111 to 1005 mg/L in 2019. However, in 2016, the chloride concentration varied between 161.99 and 5564.6 mg/L, with a maximum measured toward the south eastern part of the basin, near Oued Meliane, highlighting the significant impact of wadi wastewater on groundwater contamination (Figure 7). Taking into account the WHO-permitted chloride limits for drinking water of

250 mg/L, all groundwater samples exceed the WHO guidelines. High concentrations of chloride in drinking water can result in repeated vomiting and diarrhea [49].

Sulfate concentrations in Mornag shallow groundwater samples, which range from 65 to 1174.96 mg/L in 2013, from 61.94 to 1621.7 mg/L in 2016, and from 172.604 to 1480.166 mg/L in 2019, derive from the potential dissolution of sulfate minerals and/or the leaching of gypsum-rich marls from the Triassic and upper Oligocene formations. The lowest values are recorded in the southwest of the basin and increase to the northeast according to the direction of groundwater flow (Figure 7). Most groundwater samples are distinguished by SO$_4$ concentrations higher than the [43] standard, which is about 250 mg/L. This can result in an unpleasant taste and gastrointestinal alteration manifested by diarrhea [43].

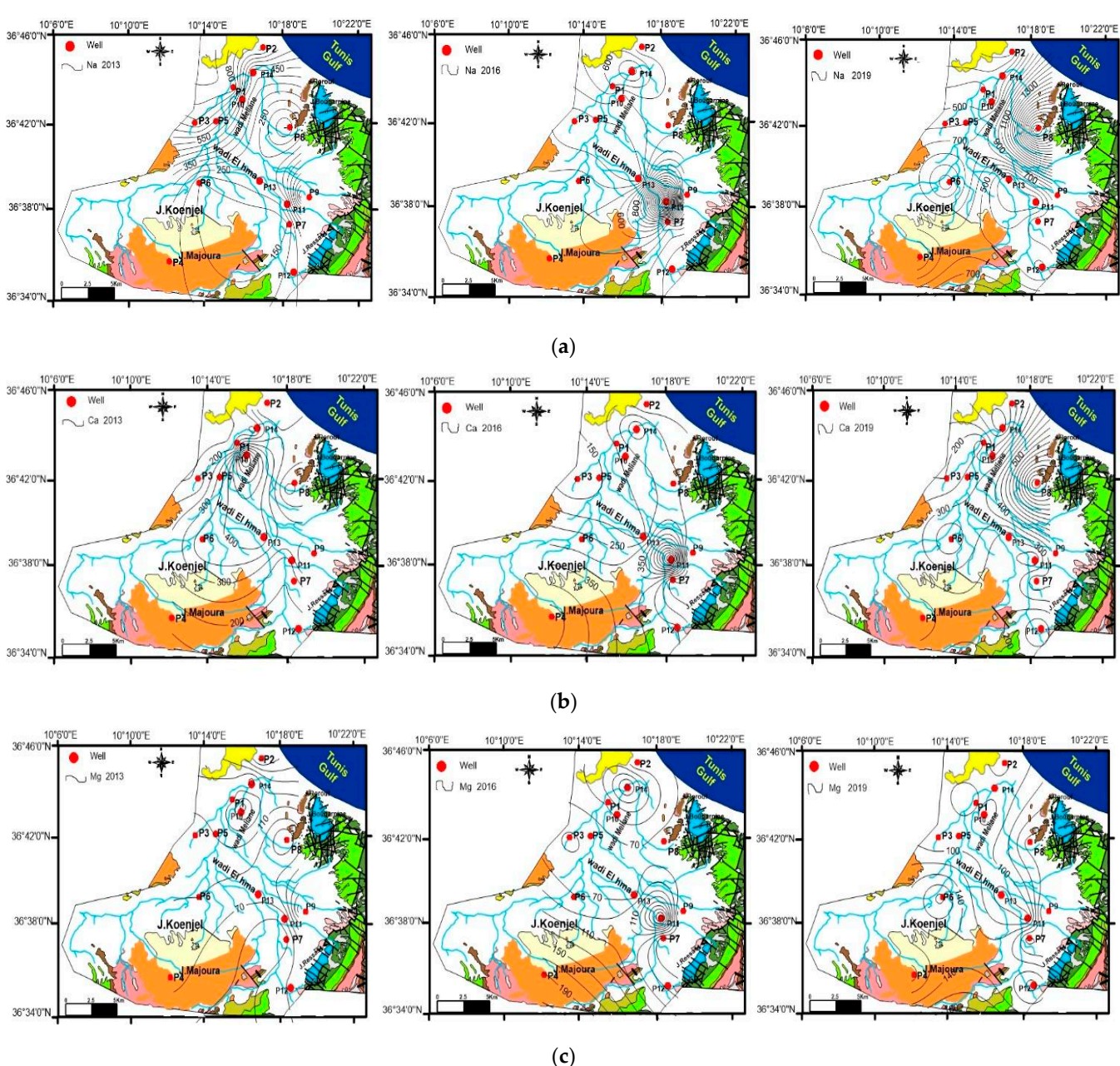

**Figure 6.** (**a**) Spatial distribution maps of cations for 2013; (**b**) spatial distribution maps of cations for 2016; (**c**) spatial distribution maps of cations for 2019.

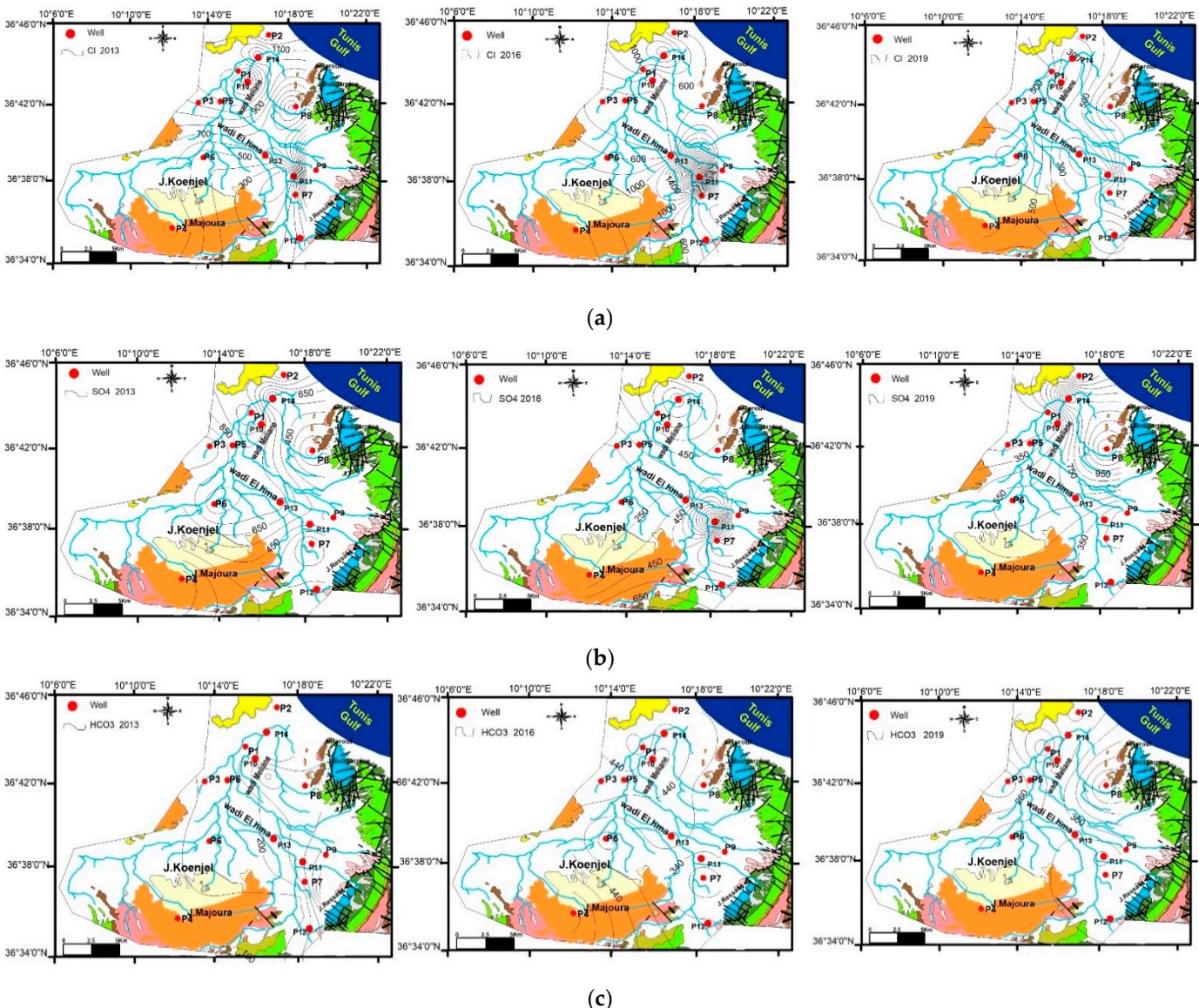

**Figure 7.** (**a**) Spatial distribution maps of anions for 2013; (**b**) spatial distribution maps of anions for 2016; (**c**) spatial distribution maps of anions for 2019.

The spatial distribution of $HCO_3^-$ displays concentrations ranging between 109.8 and 480 mg/L in 2013, from 244 to 616.1 mg/L in 2016, and from 160 to 600 mg/L in 2019. This can be related to the water–rock interaction in relation to the dissolution of relatively abundant carbonate minerals at the foot of the surrounding hills. The $HCO_3^-$ content in Mornag shallow groundwater is often higher than the standard WHO drinking water level (380 mg/L), which makes the groundwater of the study area unsuitable for consumption.

### 3.5. Groundwater Quality and Assessment for Irrigation

The suitability of groundwater irrigation was evaluated based on salinity, alkalinity, the sodium adsorption ratio (SAR), and sodium percentage (%Na) [36,37].

Indeed, high salinity of irrigation water may affect the growth of agricultural plants and soils, which reduces productivity and increases the osmotic pressure of the soil solution [50,51].

The indices of irrigation water quality in the Mornag shallow groundwater samples are given in Table 2. Irrigation water indices calculated in 2013 show that most groundwater samples (1, 2, 3, 4, 5, 9, 11, 12, and 14) are doubtful to unsuitable for irrigation; however, one

sample (10) indicates doubtful or good water for irrigation, and four samples (6, 7, 8, and 13) indicate good or excellent water class. All groundwater samples in 2016 indicate doubtful to unsuitable water quality, with only one sample (12) having a good water class. Similarly, in 2019, all groundwater samples indicate doubtful to unsuitable water quality, and two water samples (2 and 13) show a permissible to good water class. The soils in the study area are predominantly calcimagnesic soils and rendzines, while in the western and southwest parts of the study area, there is a sandy, silty texture. Under normal irrigation practices, root soils move more water out of the root zone than textured soils. As a result, sandy soils can withstand the higher salinities of irrigation waters. Dissolved salts are leached out of the root zone and may lead to sodification, the degradation of physical soil properties, the dissociation of organic matter, and the salinization of the soil. The examination of the quality indices variation over time shows an irregular pattern over the three years studied. A Wilcox diagram (Figure 8) shows that the Mornag groundwater samples were classified into three–four different categories (permissible, doubtful, good, and excellent). Indeed, the percentage of sodium (% Na) varies significantly with a minimum value in 2013 at P6 and a maximum value at P8 in 2019. For the sodium adsorption ratio (SAR) values, there is an increase that characterizes the P8 well over time, with a minimum calculated in 2013 and a maximum in 2019 (Figure 9).

The irrigation water quality index (Table 2) shows that the majority of Mornag groundwater samples have high electrical conductivity and high sodium hazard, suggesting that their quality is generally unsuitable for irrigation [2].

**Table 2.** Irrigation groundwater quality index (2013, 2016, 2019).

| Well | 2013 | | 2016 | | 2019 | |
|------|------|-----|------|-----|------|-----|
| | %Na | SAR | %Na | SAR | %Na | SAR |
| P1 | 72.4 | 65.9 | 67.8 | 48.7 | 58.2 | 35.9 |
| P2 | 56.6 | 42.1 | 66.5 | 39.4 | 50.7 | 17.9 |
| P3 | 73.6 | 63.9 | 62.5 | 25.8 | 68.2 | 49.4 |
| P4 | 50.3 | 20.2 | 42.8 | 25.8 | 53.1 | 24.2 |
| P5 | 60.9 | 25.4 | 44.7 | 27.6 | 69.6 | 81.9 |
| P6 | 23.6 | 9.3 | 58.8 | 36.2 | 65.1 | 63.4 |
| P7 | 36.2 | 12.0 | 75.2 | 53.5 | 53.4 | 36.7 |
| P8 | 29.6 | 8.4 | 59.3 | 37.2 | 70.3 | 114.7 |
| P9 | 56.1 | 42.4 | 76.5 | 55.2 | 54.3 | 37.9 |
| P10 | 34.3 | 21.7 | 53.5 | 29.0 | 54.8 | 23.2 |
| P11 | 49.6 | 22.1 | 72.4 | 119.7 | 50.5 | 19.7 |
| P12 | 48.8 | 20.3 | 32.2 | 11.8 | 65.7 | 36.8 |
| P13 | 33.7 | 16.1 | 48.8 | 20.2 | 39.8 | 15.4 |
| P14 | 66.9 | 40.9 | 62.5 | 53.6 | | |

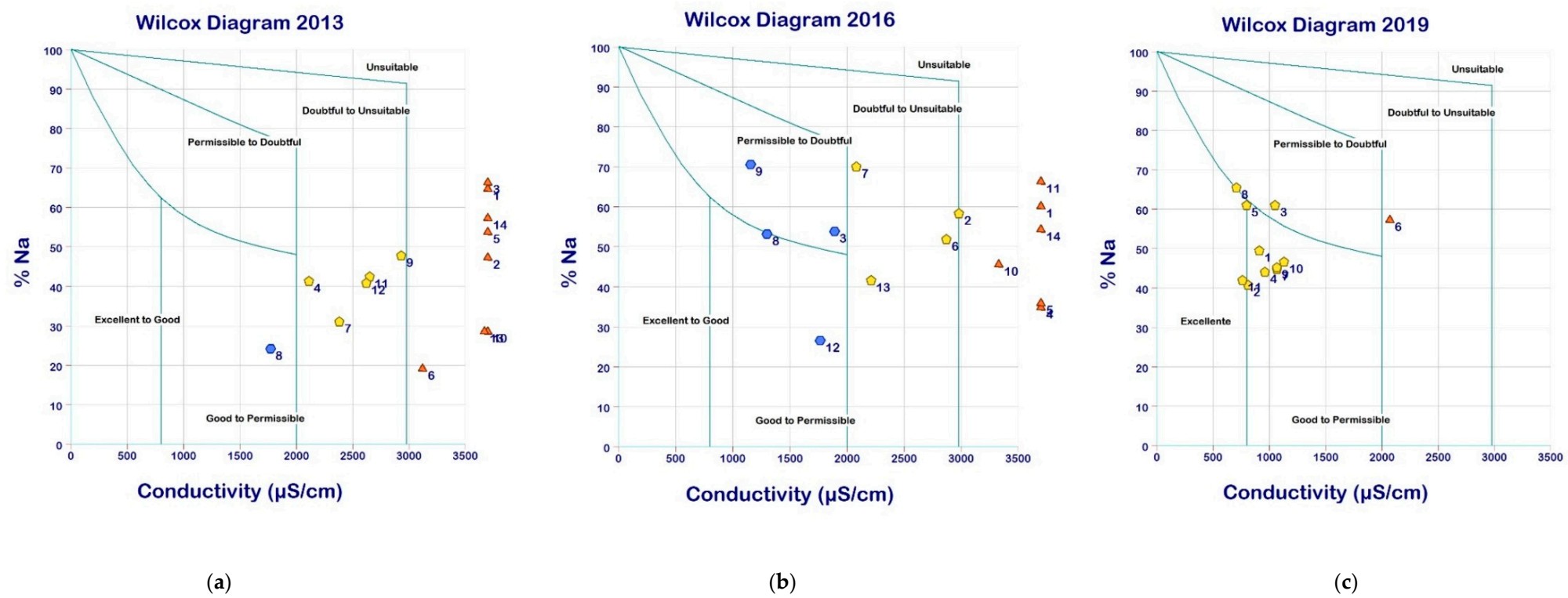

**Figure 8.** (**a**) Classification diagram of the waters of the Mornag aquifer by percentage of sodium (%Na) for 2013; (**b**) classification diagram of the waters of the Mornag aquifer by percentage of sodium (%Na) for 2016; (**c**) classification diagram of the waters of the Mornag aquifer by percentage of sodium (%Na) for 2019.

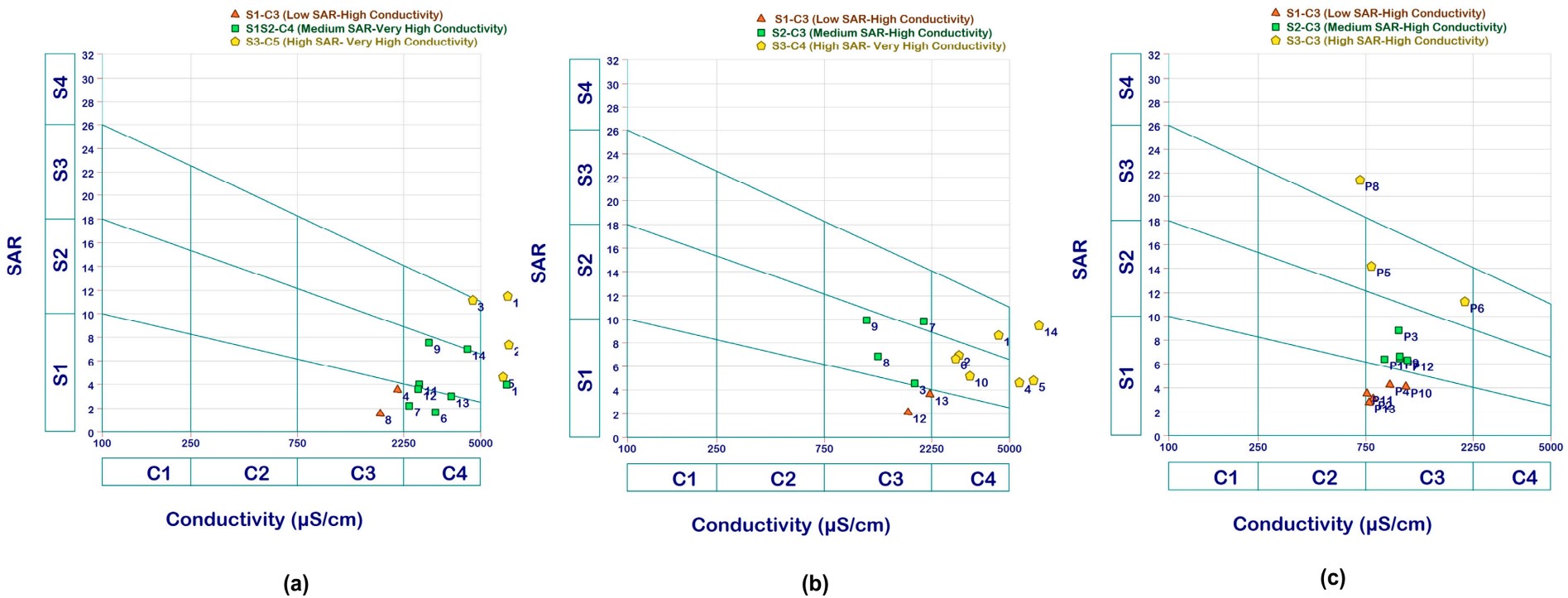

**Figure 9.** (**a**) Classification diagram of the waters of the Mornag aquifer by alkalinity hazard (SAR) for 2013; (**b**) classification diagram of the waters of the Mornag aquifer by alkalinity hazard (SAR) for 2016; (**c**) classification diagram of the waters of the Mornag aquifer by alkalinity hazard (SAR) for 2019.

## 4. Conclusions

The Mornag aquifer groundwaters flow in the South-North, Northeast-Northwest, and Southwest-Northeast directions, indicating that recharge occurs in the foothills of Khlédia, Djebel Et Tella, Er Rorouf, and Rades, while discharge occurs in the Gulf of Tunis. Recharge can be accomplished by either direct infiltration of rainwater or by linear infiltration along the beds of the wadis El Hma and Méliane.

Geochemical analysis, multivariate statistics, and water quality indices were used in the spatial–temporal evaluation of groundwater quality, identifying the main mineralization processes and determining its suitability for drinking and crop irrigation. It was revealed that groundwater quality depends largely on the geographic location and geological context of the study area. Two water facies were identified, namely, chloride–sodium–potassium and chloride–sulfate–calcium and magnesium facies, suggesting an increase in total salinity in the direction of groundwater flow. According to the calculation of saturation indices, Mornag groundwater is oversaturated with respect to carbonated minerals but undersaturated with respect to evaporative minerals.

Major ion concentrations, which increase in the direction of groundwater flow, show that water–rock interactions, wastewater recharge drained by wadi Meliane, and the return flow of irrigation water constitute the main hydrogeochemical processes controlling the composition of Mornag groundwater. The water quality indices show that all groundwater samples are classified as poor to unsuitable for drinking purposes. Furthermore, the majority of Mornag groundwater samples are not suitable for the irrigation of most soil types. In fact, SAR and EC values exhibit that high salt and sodium in the Mornag shallow groundwater may pose challenges to groundwater irrigation in the study area. Anthropogenic waste and agricultural fertilizers will continue to contaminate groundwater, leading to a deterioration in water quality. In a scenario where the quality of groundwater remains variable in time and space, a loss of cultivated soil and a decrease in agricultural production will be expected, resulting in a significant economic loss. Furthermore, intensifying climate change, marked by dry conditions, will make surface water scarce, leading to the use of groundwater for drinking water supply and irrigation. This will lead to a long-term deterioration in groundwater and soil quality and, as a result, a loss of crop yield.

**Author Contributions:** E.H., conceptualization, formal analysis, data curation, writing—original draft preparation, and writing—review and editing; A.B.M., formal analysis, writing—review and editing, and supervision; M.P., formal analysis and supervision; A.M. formal analysis and supervision. All authors have read and agreed to the published version of this manuscript.

**Funding:** Earth Sciences PhD program at Sapienza University.

**Institutional Review Board Statement:** Not applicable.

**Informed Consent Statement:** Not applicable.

**Data Availability Statement:** "MDPI Research Data Policies" at https://www.mdpi.com/ethics accessed on 26 July 2023.

**Acknowledgments:** This work has benefited from discussions with Mohamed Gharbi. The authors would like to thank him for his scientific cooperation. The authors thank the anonymous reviewers who significantly improved this paper with their valuable comments.

**Conflicts of Interest:** The authors declare no conflict of interest.

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
