# Peer review of "Spatiotemporal Changes in the Hydrochemical Characteristics and the Assessment of Groundwater Suitability for Drinking and Irrigation in the Mornag Coastal Region, Northeastern Tunisia"

_applsci, doi:10.3390/app13179887_

Round 1

Reviewer 1 Report

The paper reports an interesting hydrogeochemical approach focusing groundwater occurring in the Mornag Coastal Region, Northeastern Tunisia. The manuscript described the results obtained for a temporal hydrogeochemical monitoring held there, also highlighting changes occurring in the water quality that require the development of actions for the appropriate sustainable management of the water resources.

Author Response

We thank the Reviewer 1 for his/her appreciations. We did not provide changes, in absence of improvement request.

Best Regards

Marco Petitta

Reviewer 2 Report

This is a work on an important topic, both from a theoretical and from an applied point of view...
The territorial dimension of the problem eventually deserves a more detailed treatment. How do the chemical parameters of water vary throughout the basin and why?
The conclusion could also have a more explanatory development.
There are small typos in the text (see text) that are easily resolved...

Although my English literacy is not the best, I think the text is well written (I only detected two small problems in the conclusion part of the work)

Reviewer 3 Report

Dear authors, 

Structure of article is not upto the mark

Chemical formulas were written without considering superscript/Subscript rules. 

Headings are not appropriate like  "General settings"

references should be like [3-5] no need to write 3,4,5, and so on if the references are in series. 

Reference are very old, no reference from 2023, 2022, very less from 2015-2019 (almost no or zero) it will raise the question about advancement and novelty of the work. 

4. Results and discussion

4.1. Subsections why this heading subsections???

4.2. Figures and Tables , why this heading???

quality of figures are very very poor

Statistical analysis ; what tool aur theorem  has been used??

Discussion should be more elaborated

English is Ok

Round 2

Reviewer 3 Report

Authors have done all the revisions suggested by the reviewer

 English language fine. No issues detected